# Cadmium–Zinc Interaction in *Mus musculus* Fibroblasts

**DOI:** 10.3390/ijms231912001

**Published:** 2022-10-09

**Authors:** Ettore Priante, Edoardo Pietropoli, Elisabetta Piva, Gianfranco Santovito, Sophia Schumann, Paola Irato

**Affiliations:** 1Department of Biology, University of Padova, Via U. Bassi 58/B, 35131 Padova, Italy; 2Department of Biomedicina Comparata e Alimentazione (BCA), University of Padova, 35131 Padova, Italy

**Keywords:** cadmium, zinc, glutathione, metallothionein, fibroblast cells, MTF-1

## Abstract

This work aimed to evaluate the effects of zinc (Zn) relating to cadmium (Cd)-induced toxicity and the role played by MTF-1. This transcription factor regulates the expression of genes encoding metallothioneins (MTs), some Zn transporters and the heavy chain of γ-glutamylcysteine synthetase. For this reason, two cell lines of mouse fibroblasts were used: a wild-type strain and a knockout strain to study the effects. Cells were exposed to complete medium containing: (1) 50 μM ZnSO_4_ (Zn), (2) 1 μM CdCl_2_ (Cd 1), (3) 2 μM CdCl_2_ (Cd 2), (4) 50 μM ZnSO_4_ + 1 μM CdCl_2_ (ZnCd 1) and (5) 50 μM ZnSO_4_ + 2 μM CdCl_2_ (ZnCd 2) for 4, 18 and 24 h. Following exposure, cell viability, the intracellular content of metals, glutathione (GSH) and MT and the gene expression of the two isoforms of MT was evaluated. The results obtained suggest that a lower Cd content in the co-treatments is responsible for the protection offered by Zn due to the probable competition for a common transporter. Furthermore, Zn determines an increase in GSH in co-treatments compared to treatments with Cd alone. Finally, the MTF-1 factor is essential for the expression of MT-1 but not of MT-2 nor probably for the heavy chain of γ-glutamylcysteine synthetase.

## 1. Introduction

Cadmium (Cd) is not naturally found in the biota of organisms. Its distribution is influenced by the production and remineralization processes of biogenic particles, which are mainly released by recycled metal. After uptake by biological systems, it is subsequently discarded as detritus and faecal pellets [1]. Due to increasing extensive industrial use also in supplementation with other chemical compounds, the presence of Cd is rising [2]. Cd is highly toxic and alters, for example, antioxidant enzymes and gene expression, modifying the properties of the cell membrane and affecting channel functions. Recent studies have also considered human erythrocytes and Band 3 protein transport: Cd alters homeostasis but seems not to alter protein expression levels [3].

It is well known that heavy metals frequently interact with biological molecules. Cd has a high affinity for the sulfhydryl groups of cysteine and competes against zinc (Zn) for the structural and active sites of various enzymes, thus, impairing their catalytic activities. Cd also exerts several toxic effects on calcium (Ca) metabolism in cells. Essential trace metals, such as Zn, are required in various physiological and metabolic processes.

Zn can protect cells against oxidative damage by binding to vicinal sulfhydryl groups, preventing intramolecular disulfide formation [4,5].

Furthermore, Zn is a strong inducer of metallothionein (MT) that binds Cd and other metals. MTs are small proteins (6000–7000 daltons) rich in cysteine residues and lacking both aromatic amino acids and histidine. These proteins have a high affinity for metals and are closely associated with detoxifying toxic metal ions, such as Cd^2+^ [6,7,8,9]. Various stimuli, especially heavy metals or several stress conditions, may induce tissue accumulation. They have numerous functions, e.g., the regulation of essential metal content, the detoxification of essential and non-essential metals and the non-enzymatic scavenging of ROS [10,11].

MTs are involved in interaction mechanisms with GSH, a tripeptide with a sulfhydryl group synthesized from glutamate. GSH is present in the cell at millimolar concentrations and acts as a redox buffer to maintain the overall cellular redox state. It is the first line of defense against Cd and Cu toxicity [5,12].

The induction of MTs depends on the metal-regulatory transcription factor (MTF-1).

MTF-1 directly coordinates the regulation of genes involved in Zn homeostasis and protection against metal toxicity and regulates the well-coordinated expression of genes in response to metal ions, oxidative stress hypoxia, amino acid starvation and various stress conditions [13]. These include, but are not limited to, MT [14], zinc transporter-1 (ZnT-1) [15] and γ-glutamylcysteine synthetase heavy chain (ɣGCS_hc_) [16]. In the promoters of these genes, there are multiple copies of metal-responsive elements (MREs). MTF-1 regulates the coordinated expression of genes that present multiple copies of MRE in their promoters. Further, it functions as a Zn sensor whose DNA-binding activity is reversibly activated by intracellular free Zn pool changes. Therefore, we hypothesize that the well-known protective effect of Zn against ROS may involve the Zn-induced expression of some of these genes [17]. Cells exposed to an excess of Zn exhibit an increased expression of MT genes, which encode for the major intracellular Zn storage proteins, and ZnT-1, which excretes the metal from the cell.

In mammalian cells, two major forms, MT-1 and MT-2, are expressed. Several studies have reported that cells lacking MT-1 or MT-2 are more sensitive to a wide range of stressors, such as oxidative stress and excess heavy metals.

Given the critical role of MTF-1 in regulating several genes involved in metal homeostasis, we wanted to examine its role in the defense against Cd-induced toxicity. Since MTF-1 is also sensitive to Zn concentrations, we set up experiments in which cells were co-treated with both metals. The use of a null cell strain for MTF-1 should highlight the role played by this factor in response to Cd-induced toxicity and related defense mechanisms.

## 2. Results

In this study, we analyzed the effects of treatment with Zn 50 µM (Zn), Cd 1 µM (Cd 1), Cd 2 µM (Cd 2), Zn 50 µM + Cd 1 µM (ZnCd 1) and Zn 50 µM + Cd 2 µM (ZnCd 2) at different time points by valuing vitality cell, metal (Zn and Cd), GSH and MT content and levels of both MT-1 and MT-2 mRNAs. One of the objectives was also to clarify the antioxidant role of Zn in a particular fibroblast cell line, an SV40-transformed MTF-1 mutant that does not express the MTF-1 gene.

### 2.1. Cell Viability

Cell viability in the WT cell line remained at 100% for all three observation times in the controls and the Zn group. In the other groups, after 18 h of treatment, cell vitality decreased at least 50% compared to the level measured at 4 h. Treatments with only cadmium also decreased cell vitality after 18 h, whereas cotreated groups stagnated or increased. The treatments only treated with cadmium reached more than 80% lower cell viability than the control group. In the ZnCd 2 co-treated group, cell viability remained the same between 18 and 24 h, while in the ZnCd 1 co-treated group, there is even an improvement: viability passes from 64% at 18 h to 89% at 24 h (Figure 1).

As also seen in the WT strain, the control group and the Zn-treated group, the cell viability level remained at 100% during the experimental time. In the other groups, we observed a time-dependent cell viability reduction, higher in the two groups treated with Cd than in the corresponding co-treatments with Zn.

For both strains, no significant difference was observed after 4 h between the different treatments (Figure 1a). After 18 h in the WT strain, all four groups with Cd had significantly lower viability, by about 40% compared to both the C and the group treated with Zn. In the MTF-1 null strain, only the two groups with only Cd are significantly lower, again by about 40%, compared to the other four groups (Figure 1b). After 24 h for the WT strain, there was a statistically significant reduction in cell viability (about 80%) for treatments with Cd only and a significant reduction, equal to 40%, for the ZnCd 2 group. The MTF-1 null strain shows marked decreases in viability, around 50%, for treatments with Cd and ZnCd 2. The reduction in the ZnCd 1 group was a viability of 20% (Figure 1c).

However, for treatments with Cd 2, ZnCd 1 and ZnCd 2, the viability of the MTF-1 null strain was statistically greater than 24, 40 and 35%, respectively, compared to the WT strain.

After 24 h, the MTF-1 null strain shows marked reductions in viability, around 50%, for treatments with Cd and ZnCd 2. The decrease in the ZnCd 1 group was viability of 20%. Further, in this case, the difference in viability between the two strains for the treatment with Zn was minimal. In contrast, for the treatments with Cd alone, the MTF-1 null strain had viability more significant than 20% compared to the WT strain.

### 2.2. Accumulation of Cd

Both cell strains, control and the one treated with Zn, did not show Cd content. In the WT, cells treated with Cd 1 showed the highest value after 18 h of exposure, about four-fold, and maintained at the same level after 24 h, while MTF-1 null cells peaked after 18 h and decreased after 24 h to the level measured after 4 h (Table 1). Cd accumulation measured in cell-free extracts of cells treated with Cd 2 showed different patterns over time than Cd 1-exposed WT cells. There was a visible increase in Cd levels over 24 h. The level was significantly higher than all other levels measured after 24 h. In the MTF-1 null strain, the Cd 2-treated cells showed the same increase after 18 h and decreased after 24 h, as in the Cd 1-treated cells but were much higher than the other group (67%). In MTF-1 null cells, on the other hand, Cd levels increased over all three time points. The ZnCd 2-treated cells of both strains increased over time but showed more than twice as high Cd values in the MTF-1 strain. After 4 h in both co-treatments, the Cd content was higher than in cells only treated with Cd. After 18 h, the Cd content decreased in the ZnCd 1 group compared to Cd 1. Instead, in MTF-1 null strain, the Cd decrease in co-treatment was evident already after only 4 h and, with the higher dose (Cd 2), continued until 24 h Cd 1.

### 2.3. Accumulation of Zn

The concentration of Zn in the different treated WT cell strains showed two distinct trends. While the concentration of Zn decreased in the groups only treated with Cd, the level dropped in the co-treated groups. However, single treatment and co-treatments increased over time in the null strain. The comparison between the two strains showed significant differences in the Zn content at the different time points. A significantly lower amount of Zn was observed in the null strain after 4 h, while at 18 and 24 h, the Zn content was higher in this strain.

In the WT cell line, the Zn content after 4 h increased only in Zn, which showed values almost four-times higher than the control and ZnCd 2 groups (+46%), although in the latter, the increase is less relevant. After 18 h, the Zn content increases by about 80% in the ZnCd 1 group and after 24 h in the Cd 2 group, by more than 200%. In MTF-1 null cell line, the Zn content after 4 h increased in Zn, ZnCd 1 and ZnCd 2 groups. After 18 and 24 h, Zn content increased in the Cd 2 group (Table 1).

### 2.4. GSH Content

As already reported in the Introduction, the factor MTF-1, among the different functions, induces the expression of the gene for the catalytic subunit of glutamate-cysteine ligase (GCLC), an essential enzyme of glutathione [16]. Despite this, in the MTF-1 null strain, we found the presence of this tripeptide.

In the WT cell line, the comparison between the times within the same treatment showed, for all treatments, including control, an increase in the GSH content from 4 to 18 h. However, only in the two co-treatments with ZnCd did it continue growing, even at 24 h, a time when, in the other treatments, a marked decrease was found (Figure 2).

In control, MTF-1 null GSH content is the same at 4 h and 18 h and increases after 24 h (Figure 3).

In the Zn-treated cells, the highest content was present at 24 h, while the lowest at 18 h.

In Cd 1 and ZnCd 2 MTF-1 null-treated cells, the GSH content decreased in a time-dependent manner. In Cd 2 and ZnCd 1, the highest GSH content was present after 4 h, while the lowest was after 18 h.

After 4 h of treatment, the GSH content decreased in two Cd groups in both cell lines: about 66% in WT (Figure 2) and 30% in MTF-1 null strain (Figure 3). In other treatments, no difference was present in the WT cell line, while in MTF-1 null, there was an increase.

In the WT cell line, after 18 and 24 h of treatment, the GSH content decreased in all groups compared to the control, except for ZnCd 1.

Similar to the WT cell line, the Cd treatment caused a decrease in GSH in the MTF-1 null line. The reduction of Cd increased with exposition time. In fact, at 18 h, also ZnCd 1 had a lower content and, after 24 h, also in the other groups.

In particular, after 4 h, Zn and ZnCd 2 treatments showed an increase of 13.5% compared to the control, while in the ZnCd 1 treatment, the increase was 32%. On the other hand, treatments with Cd 1 and Cd 2 revealed a reduction of 36.5% and 29% compared to the control. After 18 h of treatment, the GSH content in the Zn treatment was the same as in the control, while it was reduced in all other treatments. The lower content is present in the treatment with Cd 2 (−81%). After 24 h, all treatments had a lower GSH content than the control, with a reduction ranging from 25% in the treatment with only Zn to 91% in both treatments with only Cd.

### 2.5. Metallothionein Content

In the WT cell line, comparing different time points showed statistical differences in all treatments, except the Cd 1 group, because there was no difference at 4 and 24 h. Generally, a sensible decrease was present at 18 h (Figure 4).

In the MTF-1 null strain, the temporal comparison showed a reduction in the MT content of the controls and treatment with Cd 1 after 18 and 24 h compared to 4 h. The treatments with Zn and the co-treatments with ZnCd indicated a gradual reduction, while treatment with Cd 2 was constant in time (Figure 5).

In the WT cell line, only after 4 h, Zn treatment induced an increase of MT (+22%), while other treatments did not have any statistical difference from control.

After 18 h, Zn and Cd 1 treatments showed no difference to control, while the other treatments showed a decrease.

After 24 h, the highest MT concentration (+104% than control) was present in the Cd 2 group and a little increase was also present in the ZnCd 1 group (Figure 4).

In the MTF-1 null cell line after 4 h, only two Cd treatments had a more outstanding MT content (+31%). After 18 h, the Cd 2 group had the highest MT concentration (+177%), while Zn and Cd 1 groups had an MT content greater than 55%. After 24 h, the highest MT content was in the Cd 2-treated cells (+152%) compared to the control. The Cd 1 group had a higher MT content of 54%, while the other treatments had a decrease of 54% (Figure 5).

Additionally, a comparison between the two strains was performed. Only after 24 h did the controls reveal differences. At the same time, the WT strain shows a higher content of MTs.

The treatments with only Zn and the co-treatments with ZnCd show a similar trend with higher values after 4 and 24 h but lower (excluding ZnCd 2) after 18 h in the WT strain. In the case of treatment with only Cd, the highest MT content is found in the MTF-1 null strain, except after 24 h, when there are no differences for Cd 1, while the values are higher for the WT strain to Cd 2.

### 2.6. MT-1 mRNA Gene Expression

In the WT strain, over time, the controls and treatment with ZnCd 2 show a progressive increase in gene expression, contrary to treatment with Zn where the growth is visible only after 24 h (+416%). In the treatment with Cd 1, there is a decrease of 43% after 24 h. In the case of treatment with Cd 2, there is an increase of about 190% after 18 h followed by a decrease to 26% after 24 h. Finally, treatment with ZnCd 1 determined an expression with a constant trend (Figure 6).

In the MTF-1 null strain (Figure 7), the gene expression remained stable over time, except for ZnCd 2, which decreased after 24 h, while Zn increased over time.

When compared to the control, the MT-1 gene expression in the WT strain after 4 h of exposure (Figure 6) is increased significantly after treatments with Cd 1 (+119%) and ZnCd 1 (+86%), while it is unchanged in treatments with Zn and Cd 2. Contrary to expectations, treatment ZnCd 2 expression appears to result in a 71% reduction compared to the control. After 18 h exposure, there was a significant increase in gene expression in Cd 1 and Cd 2 (+50 and +167% compared to control, respectively), while the others did not indicate any significant variations.

After 24 h exposure, only the Zn treatment revealed a significant (+163%) increase in gene expression. In the MTF-1 null strain (Figure 7), the MT-1 gene expression after 4 h is increased in the treatments with Zn (+116%), Cd 1 (+385%) and Cd 2 (+60%) but was unchanged in the ZnCd 1 treatment.

After 18 h of exposure, Zn and Cd 1 treatments reduced gene expression (53 and 57%, respectively), while the other treatments did not change significantly compared to the control.

After 24 h of exposure, the Zn and Cd 2 treatments led to an increased expression of 59 and 60%, respectively, while the other treatments did not involve changes.

### 2.7. MT-2 mRNA Gene Expression

As for the MTF-1 null strain, it does not express the MT-2 gene.

In the WT strain (Figure 8), the MT-2 gene expression after 4 h of exposure is, for all the treatments, lower than control values.

After 18 h, only the ZnCd 1 treatment showed an evident increase in MT-2 expression.

After 24 h, the most significant value of the expression was seen in the Zn treatment (+190%) followed by the Cd 2 treatment (+83%) compared to the control. Treatments with Cd 1 and ZnCd 1 showed a decrease in expression.

## 3. Discussion

Heavy metals may interact with proteins, inhibit enzymatic activity and promote the generation of the toxic hydroxyl radical. Homeostasis in the organism is essential to protect its health. Metal–homeostatic mechanisms include membrane transporters to balance influx and efflux, chelating molecules, such as GSH, and metal-binding proteins, such as MTs. GSH is the first line of defense before the induction of MT occurs, but it is also a substrate of some enzymes, such as glutathione peroxidase. The reduced form represents the primary intracellular pool and its concentration is much greater than that of the oxidized glutathione disulfide (GSSG). The high ratio of GSH/GSSG allows the cell to chelate excess metals.

Our results support that cells treated with Cd and Zn alone have an intracellular accumulation of the same metal compared to co-treated cells. This Zn-induced inhibition has been observed in several studies [18,19]. Zn might compete with Cd for a cellular transporter, resulting in the least intracellular accumulation. Mishima et al. [20] hypothesized that this transporter may be a voltage-gated Ca channel. However, the divalent cation transporter (DMT1) is also a known channel for transporting Cd and Zn ions.

Another possible transporter involved in the absorption of Cd was suggested by He et al. [21] by studying fetal mouse fibroblasts. It would be the ZIP8 protein, encoded by the *Slc39a8* gene, with a high affinity for this metal and, at the same time, has proven to be an essential carrier of Zn in some cell types.

Cd, therefore, appears to displace Zn from this transporter, interfering with its absorption and transport into cells [22,23].

Both tested metals interact complexly, influencing the absorption and further metabolism of the other [24]. In general, it is plausible that the lower intracellular accumulation of Cd determines the greater tolerance to Cd. In future studies, the expression of the ZIP8 transporter will be examined, but also of others, such as ZnT1.

Cd intake is connected to GSH levels and ZIP8 expression. Aiba et al. [25] showed that high levels of GSH can limit the absorption of Cd by inhibiting the Sp1 factor responsible for the transcription of ZIP8, thus, reducing the toxicity deriving from this metal. In our experiments, such an event may occur, where we observe an increase in GSH and a decrease in Cd.

According to several studies [26,27,28,29], Cd treatments involve a reduction in the content of GSH, which was also shown in our experiments. GSH decreases in two ways: on the one hand, it detoxifies Cd directly [30], binding it to the six different bond sites; on the other hand, it acts as a scavenger for the radical species that Cd indirectly produces. Homeostasis is altered and the cells can no longer reduce GSSG to GSH to maintain oxidative balance and export it from the cell. Given that an enzymatic reduction in GSSG restores GSH, the progressive exhaustion of the latter goes hand in hand with the decrease in the concentration of GSH [31].

The protective effect of Zn against Cd-induced toxicity can also be explained by the fact that Zn can increase the GSH content and reduce its oxidation through indirect mechanisms [27,32].

In the case of reductions in GSH content following exposure to Zn, Walther et al. [33] also observed this, which corresponded to significant reductions in the GSH/GSSG ratio. Therefore, the GSH content may decrease, although cell viability is not compromised.

Comparison between the two cell strains highlights the overall higher GSH content in the MTF-1 null strain, with few exceptions. This explanation would seem surprising, considering that the MTF-1 factor modulates the expression of the heavy chain of y-glutamylcysteine synthetase, an enzyme involved in synthesizing GSH. Instead, according to Wimmer et al. [34], the *Gclc* gene coding for y-glutamylcysteine synthetase is induced by Cd through a path independent of that which involves the activation of MTF-1. Therefore, it would appear that the MTF-1 null cells can fully exploit this alternative mechanism to counteract Cd-induced oxidative stress.

In ZnCd co-treated cells, the Zn content, higher than in control, is, therefore, crucial in buffering the reduction in GSH with the mechanisms mentioned above.

Regarding the induction of MTs, in the WT strain, there is a very evident induction, only in the treatment with Cd 2 at 24 h, which, however, has the highest content of intracellular Cd. Treatments with Cd alone determine, at least in the MTF-1 null strain, a robust induction of MTs that occurs after 4 h, when metal accumulation is still low. One possibility is that the two strains have a threshold dose of activation of MT synthesis, which is lower in the MTF-1 null strain, while it is reached only with the highest amount of Cd and after 24 h for the WT strain. One of the feasible induction mechanisms involves the substitution of Zn by Cd from the cysteine sites of MT due to the higher affinity that Cd has for them (the affinity scale of metals for MT is Fe (II) ≈ Zn (II) ≈ Co (II) < Pb (II) < Cd (II) < Cu (I) < Au (I) ≈ Ag (I) < Hg (II) < Bi (III)). Free Zn, therefore, induces the MT synthesis by saturating the sites for the Zn of the factor MTF-1, which can then bind to DNA and proceed with transcription. However, the lack of MTF-1 in the MTF-1 null strain should prevent this strain from synthesizing MT. Only the expression of the MT-2 isoform is not induced. Since this strain, on the other hand, expresses the MT-1 isoform, there must be a mechanism independent of the one involving the MTF-1 factor for the induction of MT synthesis. Several studies have confirmed the existence of an alternative tool that uses the bHLH-Zip protein, better known as the USF factor (upstream regulatory factor), consisting of three isoforms [35]. The promoter region of the MT-1 gene contains ARE (antioxidant response element) sequences that overlap a site USF binder. The USF/ARE complex can increase the expression of the mouse MT-1 gene when induced by Cd and hydrogen peroxide (H_2_O_2_) but not by Zn [36]. This mechanism consequently seems efficient in ensuring the production of the MT-1 isoform by compensating for the lack of the MT-2 isoform, given that at 4 and 18 h, for treatments with Cd only, the MT content in the MTF-1 null strain is higher than that of the WT strain.

We do not observe the synergism of Cd and Zn for the MT induction and this could be due to the general lower accumulation of intracellular Cd compared to treatments with Cd alone. Zn probably prevents the accumulation of Cd, resulting in a lower induction of MT in co-treatments, occurring to a greater extent in the MTF-1 null strain, lacking the induction mechanism through MTF-1. Therefore, as Mishima et al. [20] proposed, the protection against Cd-induced toxicity that Zn offers could take place not involving MT but is based on the minor accumulation of Cd caused by Zn. Thereby, Zn competes at the level of the intake sites. It would act as a regulator of intra- and extracellular Cd through the similar chemical properties that allow it to mimic.

As for gene expression, in the controls of the WT strain, the mRNA, especially for isoform 2, has a time course like the MT content shown. Concerning treatment with Zn alone, a considerable increase in gene expression is observed in both isoforms only after 24 h of treatment. However, it does not reflect the rise in the MT content. This discrepancy is likely due to the different half-life of the mRNA compared to that of the protein.

However, Andrews et al. [36] observed that the exposure to Zn of murine Hepa cells leads to a peak in the expression of MT-1 after 1 h, which then returns to baseline after 9 h. Further studies on treatment times exceeding 24 h are necessary to verify whether there is an actual increase in the MT content.

As for the treatment with Cd 2, isoform 1 seems to have a shorter induction time than isoform 2: the first has an induction peak at 18 h, while the second after 24 h. However, high mRNA content for both isoforms at 24 h agrees with the higher protein content. The treatment with Cd 2 induces a higher range of MT and mRNA could be related to the increased intracellular Zn content found, given that this treatment has a quantity of metal equal to that present in the treatment with only Zn.

For the treatment with ZnCd 1, isoform 2 shows a high expression after 18 h, when protein content is lower. Again, the difference in half-life times between mRNA and proteins may explain this result: the increased expression at 18 h is reflected by an increase in MT content that occurs only after 24 h.

In the MTF-1 null strain, treatment with Zn alone causes a reduction in gene expression after 18 h, which increases again after 24 h. A reduction in the MT content accompanies a decrease between 4 and 18 h. At the same time, the subsequent increase has no comparison with the protein content: also, in this case, the different half-life times can explain these differences.

In general, gene expression trends do not always correspond to the MT content and, therefore, require further studies. Santon et al. [37] showed, with studies on murine fetal fibroblasts, that the mRNA levels determined by an RT-PCR analysis following exposure to metals do not always reflect the MT content and hypothesize that this is due to the different induction and degradation times of the metal–MT complexes.

Another explanation is given by studies on protists exposed to Cd, which have shown a continuous alternation in phases in which the expression of genes for MTs induced. Further, they indicated stages in which this expression is reduced following the intervention of mechanisms of down-regulation [38]. Further, Boldrin et al. [39] found inconsistencies between the mRNA levels of the two MT isoforms and the amount of protein, hypothesizing the intervention of post-transcription mechanisms in regulating MT induction.

It must also be considered that the induction times of MTs may vary according to the tissue and the type of metal used.

## 4. Materials and Methods

### 4.1. Cell Culture

Dko7 cells were derived from MTF-1−/− mouse (*Mus musculus* sp.) embryonic stem cells that were allowed to differentiate into fibroblast-type cells and immortalized by transformation with simian virus 40 (SV40), kindly provided by Dr Glen K. Andrews, Department of Biochemical and Molecular Biology, Medical Centre of Kansas University. Cells were grown in Dulbecco’s modified Eagle’s medium (DMEM) supplemented with 10% (*v*/*v*) fetal bovine serum, sodium carbonate (7.5%), sodium pyruvate (100 mM) and antibiotics (penicillin 100 U/mL and streptomycin 100 μg/mL) at 37 °C in an atmosphere of 5% CO_2_.

### 4.2. Treatment with Heavy Metals

For heavy metal treatment, cells were exposed to complete medium containing: (1) 50 μM ZnSO_4_ (Zn), (2) 1 μM CdCl_2_ (Cd 1), (3) 2 μM CdCl_2_ (Cd 2), (4) 50 μM ZnSO_4_ + 1 μM CdCl_2_ (ZnCd 1) and (5) 50 μM ZnSO_4_ + 2 μM CdCl_2_ (ZnCd 2) for 4, 18 and 24 h. These concentrations were estimated with the MTT cell proliferation assay based on the cellular conversion of a tetrazolium salt into a formazan product [40]. The conversion occurs only in living cells and the amount of formazan produced is proportional to the number of living cells present.

### 4.3. Determination of Metals and MT Contents

Cells at 75% confluency were washed twice with phosphate-buffered saline (PBS) and harvested using trypsin (0.25%)-EDTA (1 mM). Cells were sedimented by centrifugation at 180× *g* for 5 min, washed twice with PBS and re-suspended in 20 mM Tris–HCl buffer (pH 7.5) supplemented with 0.006 mM leupeptin, 0.5 mM phenylmethylsulphonylfluoride (PMSF) and 0.01% β-mercaptoethanol. Subsequently, cells were sonicated twice for 30 s using a Labsonic U instrument on ice. Homogenates were centrifuged at 13,000× *g* for 60 min at 4 °C. Resulting supernatants were used for metal and MT quantification. For heavy metal determinations, samples of the resulting supernatants were digested with AristaR nitric acid and Cd and Zn contents were determined by atomic absorption flame with an air–acetylene flame (Perkin-Elmer model 4000). The instruments for metal analysis were calibrated by standard addition methods and by reference to the fresh standard salt solution. Values were expressed as μg of single metal/mg of total proteins assayed by the Folin phenol reagent method [41] using bovine serum albumin as standard.

The silver saturation method determined MT concentrations in the supernatant [42]. As previously described, the amount of MTs was normalized against total soluble cell proteins.

### 4.4. Glutathione Quantification

Total glutathione (GSH + GSSG) has been determined by enzymatic recycling following the Anderson method [43], which is based on GSH oxidation by 5,5-dithiobis (2-nitrobenzoic acid) (DTNB) to give GSSG with the stoichiometric formation of 5-thio-2-nitrobenzoic acid (TNB). GSSG is reduced to GSH by the action of the highly specific glutathione reductase (GSSG reductase) and NADPH. The rate of TNB formation is recorded at 412 nm and is proportional to the nmoles of GSH and GSSG present in the sample. This rate has been compared to a GSH standard curve. Values are expressed as nmoles/g wet weight.

### 4.5. qRT-PCR Analysis

To estimate the expression of MT-1 and MT-2 mRNA, real-time qRT-PCR was performed and RNA from pooled cell samples was isolated using Biolzol reagent (Bioer Technology, Hangzhou, China). The amount of RNA was quantified using NanoDrop ND-1000 spectrophotometer (Thermo Fisher Scientific, Waltham, MA, USA), determining the UV absorbance at 260 nm. Further purity was tested using the absorbance ratio at A_260_/A_280_ nm.

First-strand cDNA was synthesized with 1 μg total RNA, 1000 ng/mL oligo (dT)_18_, 10 U Improm II Reverse transcriptase and 20 U RNase inhibitor in a final 20 μL reaction mixture containing 5× reverse transcriptase buffer, 25 mM MgCl_2_ and 10 mM dNTP. The reaction mixture was incubated at 25 °C for 5 min, at 42 °C for 1 h and 70 °C for 15 min.

PCR was carried out in a 50 μL reaction mixture containing 10 mM dNTP, 10 μM of each sense and antisense primer and 2.5 U of Taq DNA polymerase and 2 μL of cDNA. The sequence of sense and antisense primers for mouse MT-1 was TCTCGGAATGGACCCCAACTG and TTTACACGTGGTGGCAGCGC, respectively, for mouse MT-2 were CGATCTCTCGTCGATCTTCA and GGAGAACGAGTCAGGGTTGT, and for β-actin were CCAGGGTGTGATGGTGGGAATG and CGCACGATTTCCCTCTCAGCTG, respectively. After an initial denaturation at 95 °C for 2 min, amplification was carried out for 25 cycles of 30 s at 95 °C for denaturation, 30 s at 95 °C for annealing and 60 s at 72 °C for extension with a final extension step at 72 °C for 5 min. The PCR products were analyzed by electrophoresis using a 2.0% (*w/v*) agarose gel, stained with ethidium bromide and visualized under UV light. MT-1 and MT-2 mRNA levels were determined by RT-PCR analysis followed by densitometry scanning. All MT-1 and MT-2 RT-PCR products were normalized to the corresponding β-actin RT-PCR as a housekeeping gene.

### 4.6. Statistical Analysis

The results are expressed as mean ± standard deviation (SD).

The data were statistically analyzed using the PRIMER statistical software. Analysis of variance (ANOVA) followed by post hoc comparison with the Student–Newman–Keuls test was conducted to determine intraspecific differences between groups. When results were obtained only from two groups, a comparison was carried out by a two-tailed Student’s *t*-test. The significance level for all the statistical comparisons was chosen to be 0.05.

## 5. Conclusions

From the results obtained, we can conclude the following:In co-treatments, both Cd and Zn are accumulated in less quantity and this determines the greater tolerance to Cd-induced cytotoxicity.In the WT strain in the treatment with Zn alone, there does not appear to be an increase in GSH content, which probably occurs only when cells are subjected to oxidative stress. In contrast, treatments with Cd alone cause a reduction in GSH due to the probable oxidative stress that this metal entails. Instead, in co-treatments, Zn causes an increase in the GSH content compared to treatments with Cd alone, promoting its expression and preventing oxidation through indirect mechanisms, demonstrating its protective role.The MTF-1 null strain, although lacking the MTF-1 factor, still shows a high GSH content, generally higher than that of the WT strain, thanks to an alternative mechanism of expression of y-glutamylcysteine synthetase, independent of MTF-1.The two strains have different threshold doses of activation of MT production: lower for the MTF-1 null strain and higher for the WT strain.The MTF-1 null strain expresses only isoform 1 of the MT through the USF/ARE sequences, making up for the lack of isoform 2 and ensuring a higher protein production than that of the WT strain for treatments with Cd only.The time courses of the mRNAs of the MT-1 and MT-2 genes and those of the protein found are variable, caused by the different half-lives of the messenger and protein and/or by the intervention of down-regulation mechanisms of gene expression.Although the MTF-1 null strain in the literature is considered more sensitive than the WT strain, it generally exhibits greater viability following Cd-only treatments than the WT strain, possibly due to the more significant amount of MT it presents.

## Figures and Tables

**Figure 1 ijms-23-12001-f001:**
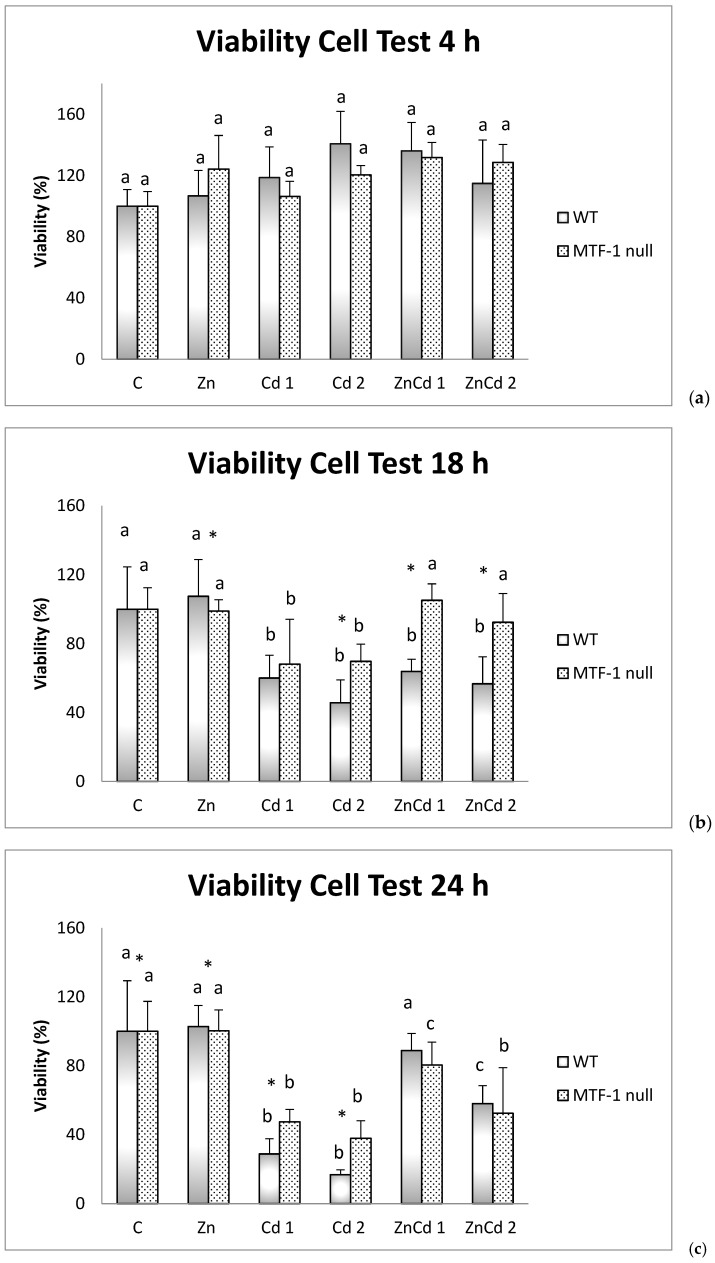
Cell viability test. The viability values are indicated as a percentage and the control is set at 100%. n = 6. (**a**) 4 h; (**b**) 18 h; (**c**) 24 h. Different letters indicate statistically significant differences between the treatments of the same strain; asterisks indicate statistically significant differences between the two cell strains. C = Control; Zn = Zn 50 µM; Cd 1 = Cd 1 µM; Cd 2 = Cd 2 µM; ZnCd 1 = Zn 50 µM + Cd 1 µM; ZnCd 2 = Zn 50 µM + Cd 2 µM.

**Figure 2 ijms-23-12001-f002:**
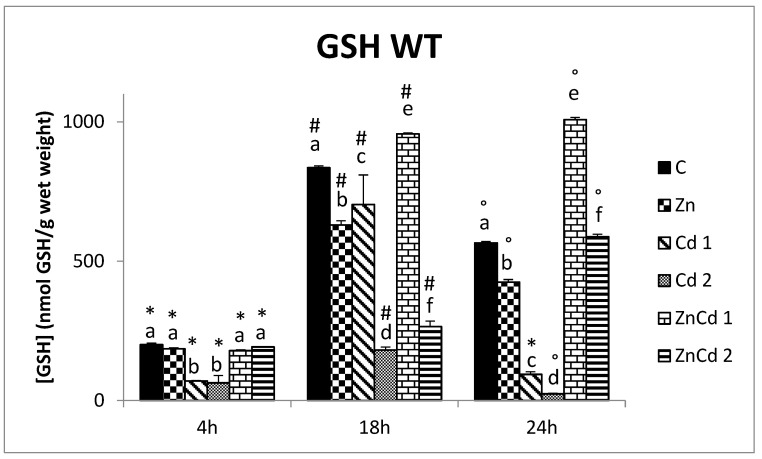
GSH content in nmoles of GSH/g of wet weight versus time in the WT strain. Different letters indicate statistically significant differences between the various treatments of the same time, while the symbols indicate statistically significant differences between times within the same treatment. C = Control; Zn = Zn 50 µM; Cd 1 = Cd 1 µM; Cd 2 = Cd 2 µM; ZnCd 1 = Zn 50 µM + Cd 1 µM; ZnCd 2 = Zn 50 µM + Cd 2 µM.

**Figure 3 ijms-23-12001-f003:**
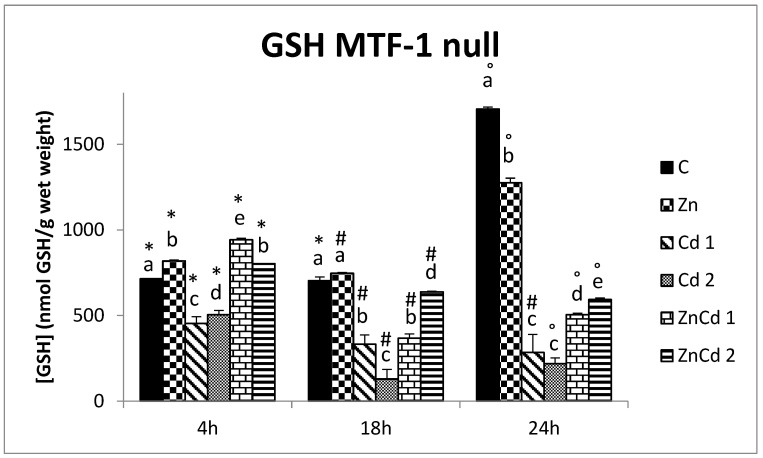
GSH content in nmoles of GSH/g of wet weight versus time in the MTF-1 null strain. Different letters indicate statistically significant differences between the various treatments of the same time, while the symbols indicate statistically significant differences between times within the same treatment. C = Control; Zn = Zn 50 µM; Cd 1 = Cd 1 µM; Cd 2 = Cd 2 µM; ZnCd 1 = Zn 50 µM + Cd 1 µM; ZnCd 2 = Zn 50 µM + Cd 2 µM.

**Figure 4 ijms-23-12001-f004:**
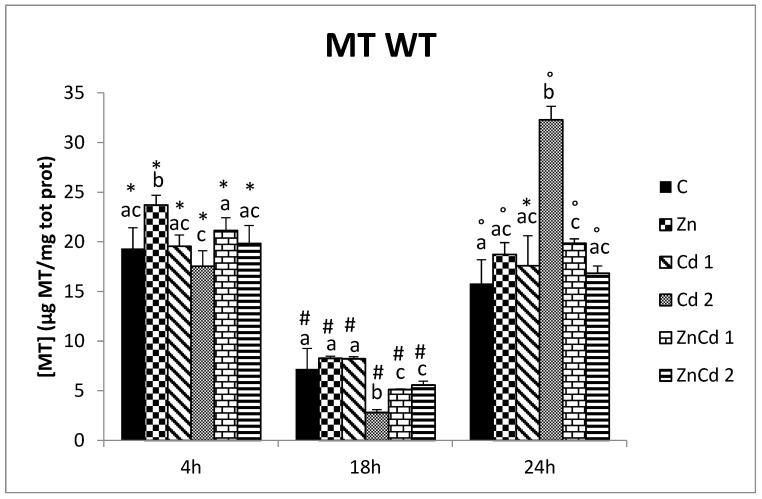
MT content in µg/mg protein versus time in the WT strain. Different letters indicate statistically significant differences between the various treatments of the same time, while the symbols indicate statistically significant differences between times within the same treatment. C = Control; Zn = Zn 50 µM; Cd 1 = Cd 1 µM; Cd 2 = Cd 2 µM; ZnCd 1 = Zn 50 µM + Cd 1 µM; ZnCd 2 = Zn 50 µM + Cd 2 µM.

**Figure 5 ijms-23-12001-f005:**
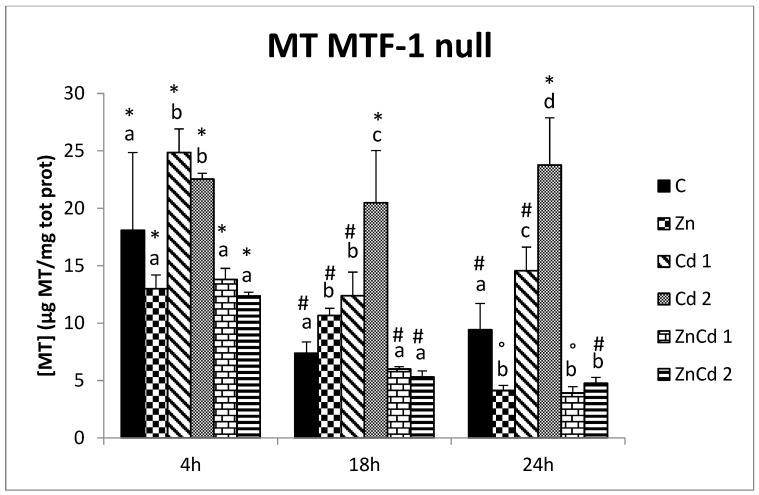
MT content in µg/mg protein versus time in the MTF-1null strain. Different letters indicate statistically significant differences between the various treatments of the same time, while the symbols indicate statistically significant differences between times within the same treatment. C = Control; Zn = Zn 50 µM; Cd 1 = Cd 1 µM; Cd 2 = Cd 2 µM; ZnCd 1 = Zn 50 µM + Cd 1 µM; ZnCd 2 = Zn 50 µM + Cd 2 µM.

**Figure 6 ijms-23-12001-f006:**
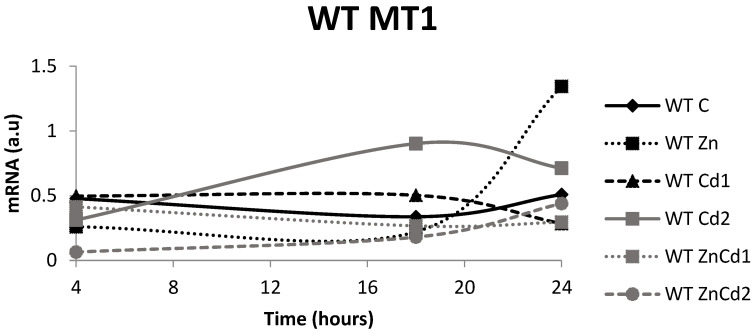
mRNA in arbitrary units (a.u.) of the MT-1 gene in the WT strain. C = Control; Zn = Zn 50 µM; Cd 1 = Cd 1 µM; Cd 2 = Cd 2 µM; ZnCd 1 = Zn 50 µM + Cd 1 µM; ZnCd 2 = Zn 50 µM + Cd 2 µM.

**Figure 7 ijms-23-12001-f007:**
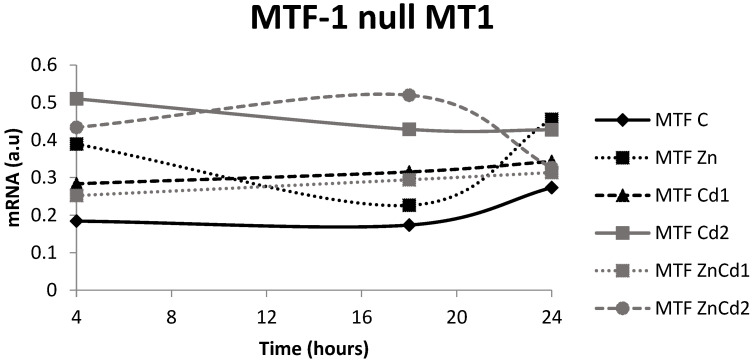
mRNA in arbitrary units (a.u.) of the MT-1 gene in the MTF-1 null strain. C = Control; Zn = Zn 50 µM; Cd 1 = Cd 1 µM; Cd 2 = Cd 2 µM; ZnCd 1 = Zn 50 µM + Cd 1 µM; ZnCd 2 = Zn 50 µM + Cd 2 µM.

**Figure 8 ijms-23-12001-f008:**
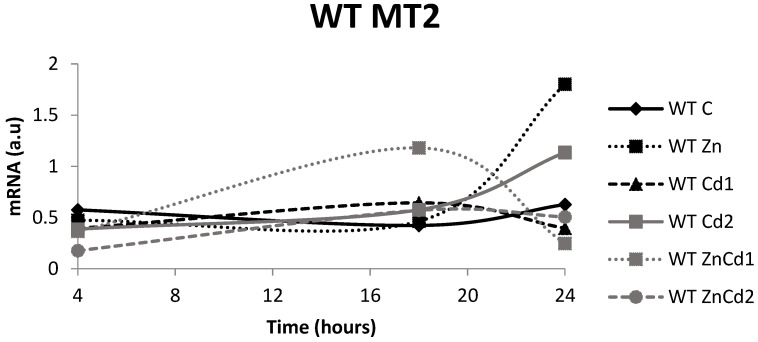
mRNA in arbitrary units (a.u.) of the MT-2 gene in the WT strain. C = Control; Zn = Zn 50 µM; Cd 1 = Cd 1 µM; Cd 2 = Cd 2 µM; ZnCd 1 = Zn 50 µM + Cd 1 µM; ZnCd 2 = Zn 50 µM + Cd 2 µM.

**Table 1 ijms-23-12001-t001:** Cd and Zn content (µg/mg protein) for the various treatments and times. Different letters indicate statistically significant differences between the various treatments of the same time, while the symbols indicate statistically significant differences between times within the same treatment. C = Control; Zn = Zn 50 µM; Cd 1 = Cd 1 µM; Cd 2 = Cd 2 µM; ZnCd 1 = Zn 50 µM + Cd 1 µM; ZnCd 2 = Zn 50 µM + Cd 2 µM.

		4 h	18 h	24 h
**µg Cd/mg protein**				
**WT**	Cd 1	0.002 ± 0.001 a*	0.033 ± 0.003 a#	0.029 ± 0.005 a#
	Cd 2	0.004 ± 0.0001 a*	0.006 ± 0.0018 b#	0.168 ± 0.0077 b°
	ZnCd 1	0.027 ± 0.0013 b*	0.018 ± 0.0002 c#	0.021 ± 0.0002 a°
	ZnCd 2	0.029 ± 0.0023 b*	0.031 ± 0.0003 d*	0.038 ± 0.0008 c#
**MTF-1 null**	Cd 1	0.021 ± 0.0016 a*	0.090 ± 0.0059 a#	0.022 ± 0.001 a*
	Cd 2	0.030 ± 0.0003 b*	0.266 ± 0.0479 b#	0.088 ± 0.0141 b*
	ZnCd 1	0.013 ± 0.0001 c*	0.023 ± 0.0005 c#	0.055 ± 0.0012 c°
	ZnCd 2	0.014 ± 0.0001 c*	0.018 ± 0.0008 c#	0.051 ± 0.0017 c°
**µg Zn/mg protein**				
**WT**	C	0.027 ± 0.0049 a*	0.024 ± 0.0119 a*	0.022 ± 0.0022 a*
	Zn	0.105 ± 0.0053 b*	0.050 ± 0.001 b#	0.066 ± 0.0038 b°
	Cd 1	0.023 ± 0.0008 a*	0.017 ± 0.0002 a#	0.023 ± 0.0038 a°
	Cd 2	0.025 ± 0.0003 a*	0.016 ± 0.0004 a#	0.065 ± 0.003 b°
	ZnCd 1	0.030 ± 0.0014 a*	0.040 ± 0.0004 b#	0.054 ± 0.0006 c°
	ZnCd 2	0.039 ± 0.0031 c*	0.040 ± 0.0004 b*	0.057 ± 0.0011 c°
**MTF-1 null**	C	0.024 ± 0.0104 a*	0.024 ± 0.0064 a*	0.028 ± 0.0015 a*
	Zn	0.044 ± 0.0005 b*	0.099 ± 0.0046 b#	0.080 ± 0.0038 b°
	Cd 1	0.026 ± 0.002 a*	0.027 ± 0.0018 a*	0.024 ± 0.001 a*
	Cd 2	0.023 ± 0.0002 a*	0.067 ± 0.012 c#	0.040 ± 0.0063 c°
	ZnCd 1	0.031 ± 0.0003 a*	0.039 ± 0.0008 d#	0.089 ± 0.002 d°
	ZnCd 2	0.032 ± 0.0002 a*	0.041 ± 0.0017 d#	0.063 ± 0.0021 e°

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
