# Peer review of "Cadmium–Zinc Interaction in *Mus musculus* Fibroblasts"

_ijms, 2022, doi:10.3390/ijms231912001_

Round 1

Reviewer 1 Report

The paper by Priante et al. reports on Cd and Zn ion uptake in reference cells and in MTRF-1 knock-out cells. They look at accumulation of metals in cells depending on metals supplied (also co-supplements with Zn/Cd), mRNA levels, GSH/GSSG and metallothionein levels etc.) This is a very exhaustive study that could potentially  deliver much insight into the effects of the heavy metals. I feel that it is of interest and should be published.

However, I have some serious problems with the paper in its present form. First of all, a native-English speaker should once go over the paper and correct for language, sometimes it was even not clear to me what the authors wanted to say.

I also find the discussion very long, poorly structured and therefore difficult to read. I suggest to first develop some mechanistic picture of what happens in the cell when these metal ions enter the cells, and the discuss the observations and conclude what has been learned from the study. The authors are clearly competent in the field, but the extend of details in the discussion makes it very difficult to read for the non-experts in the fields.

A more specific points:

I am not sure how the data are derived, and what the reported errors really mean: Sometimes trends are very astonishing. One example is Table 1: The value “jumps” from 0.006 to 0.168. I really wonder whether the 0.168 is a typo, or whether something “weird” happened to the cells. Are those biological triplicates? If not, what does the deviation really report? Are all those trends significant? I also find no explanation of what the letters and symbols In Table 1 exactly mean, I have wondered about this for quite some time..

Presentations of data vary: Sometimes lines are used, sometimes column-type presentations, sometimes data are reported in Tables. I suggest to decide for a common format. Table 1 would also be easier to grasp from a graphical representation.

I would also suggest the same scaling of the y-axis for WT and MTF-1 null data so that values can be more easily compared.

Line 107/108: How is it possible that a decrease of 150 % is observed. My simple mathematical understanding is that one then yields negative numbers…or how is that meant? (In the table the decrease is about 67 %).

In Fig. 6 and 7 curved trend lines are drawn. How is that possible without any model? How are these lines described mathematically? For me it looks like that many other shapes would be equally probable (unless there is some model behind?)

I really like the bulleted conclusions, this makes the data a bit more compact, and the major findings easier to grasp. I would really suggest to rewrite the discussion to become less detailed and explaining more the important general trends.

Author Response

Thanks for the suggestions. The manuscript was revised with respect to the English language and the discussion was shortened

A more specific points:

I am not sure how the data are derived, and what the reported errors really mean: Sometimes trends are very astonishing. One example is Table 1: The value “jumps” from 0.006 to 0.168. I really wonder whether the 0.168 is a typo, or whether something “weird” happened to the cells. Are those biological triplicates? If not, what does the deviation really report? Are all those trends significant? I also find no explanation of what the letters and symbols In Table 1 exactly mean, I have wondered about this for quite some time.

Reply:

The data reported in Table 1 are the mean values +/- the standard deviation of biological triplicates. The values are correct. The different letters indicate statistically significant differences between the different treatments (within the same time). For example, let's consider the Cd content in the WT strain: at 4h the two Cd groups (1 and 2) are not different from each other (letter a), as are the 2 co-treatments ZnCd1 and ZnCd2 (letter b, while there is a difference between single treatments (Cd1 and 2) and co-treatments (ZnCd 1 and 2). At 18 h all treatments are different from each other, and therefore all have different letters. At 24h Cd1 is not different from ZnCd1 (letter a), while they are both towards Cd2 (letter b) and towards ZnCd2 (letter c) and the latter between them. The different symbols represent statistically significant differences between the times within the same treatment, so for example, first row WT strain group Cd 1, the 4 h are different at both 18 and 24 h, while these are not between them.

Presentations of data vary: Sometimes lines are used, sometimes column-type presentations, sometimes data are reported in Tables. I suggest to decide for a common format. Table 1 would also be easier to grasp from a graphical representation.

Reply:

The choice of how to report the data was made based on how it seemed easier and more appropriate to visualize and interpret the data. In any case, we have decided to change the representation of the cell viability data, to also report the statistical significance.

I would also suggest the same scaling of the y-axis for WT and MTF-1 null data so that values can be more easily compared.

Reply:

Thanks for the suggestion. We have corrected figures 2-7, so that there is the same scaling of the y-axis for WT and MTF-1 null data.

Line 107/108: How is it possible that a decrease of 150 % is observed. My simple mathematical understanding is that one then yields negative numbers…or how is that meant? (In the table the decrease is about 67 %).

Reply:

You are right, I apologize for the mistake

In Fig. 6 and 7 curved trend lines are drawn. How is that possible without any model? How are these lines described mathematically? For me it looks like that many other shapes would be equally probable (unless there is some model behind?)

Reply:

We simply reported the data by choosing the "dispersion with curved lines and indicators" model on the Excel program.

I really like the bulleted conclusions, this makes the data a bit more compact, and the major findings easier to grasp. I would really suggest to rewrite the discussion to become less detailed and explaining more the important general trends.

Reply:

We have reviewed the discussion and the conclusions, trying to explain the most important points

Reviewer 2 Report

The authors investigated the influence on Cd toxicity of simultaneous treatment with Zn in mouse fibroblast Dko7 cells, the effect of the knockout of MTF-1 on the interaction between Cd and Zn, and the mechanisms underlying the interaction. In my opinion, the interaction among Cd and other heavy metals involvement in Cd toxicity is important for understanding the toxicity more realistically, the results in the present study are interesting. However, the manuscript and data of the results are not adequately organized, and the discussion seems to be a repetition of the results. I think, the present study should be re-organized with a focus on (1) the influence on Cd toxicity of simultaneous treatment with Zn, (2) the effect of the knockout of MTF-1 on the Cd-Zn interaction, and (3) whether intracellular accumulation of Cd and Zn, GSH, and MT-1/MT-2 are involved in the mechanisms. The current manuscript is just a list of the results, which are not well related, and some parts of the manuscript have been derailed and not sufficiently elaborated. The discussion should focus on the Cd toxicity. The overall structure of the paper should be reconsidered. I therefore recommend a reject of the paper at this time.

1. The reason for using MTF-1KO cells in the Cd-Zn interaction should be described in more detail. The author set out to compare among the contribution of MTs, GSH, and Cd accumulation in the defense of Zn against Cd toxicity in the fibroblasts suppressed the synthesis of MTs by knocked out MTF-1? If so, it is essential to examine the expression Cd transporter (e.g. ZIP8).

2. I interpreted “ZnCd 1” to mean “50 µM Zn and 2 µM Cd.” However, the such abbreviations should be mentioned in the manuscript and in the figures, respectively. The authors should describe them so that the reader does not misunderstand.

3. In Figure 1, it is common to describe it as “Viability” instead of “Vitality”. In addition, these data must be analyzed for statistical significance. In particular, as in Table 1, three major points should be analyzed: First, in order to clarify the effect of zinc on the Cd toxicity, the significant difference between the viability of Cd1 and ZnCd1, Cd2 and ZnCd2 in WT should be analyzed. Second, to clarify the effect of MTF-1 KO on its Cd toxicity, the significant difference between the viability of Cd 1 in WT and MTF-1 KO should be analyzed. Third, the significant difference between the viability of CdZn1 and CdZn2 in WT and MTF-1 KO should be analyzed to clarify the effect of MTF-1 KO on the Cd-Zn interaction. The data should also be redesigned to facilitate the understanding of these analyses. The markers in the figures are all circle-like, and there is no uniformity in the colors of the markers and lines, which will confuse the reader as it is.

4. Since it is difficult to understand what was obtained in the present study, the structure of the entire text in Discussion section should be reconsidered. In particular, it should be clearly stated whether zinc protects against Cd toxicity, what the mechanism is, and what was obtained by using MTF-1 KO. Currently, the present study only compares with previous papers, and although the validity of each and every data is discussed, it is difficult to find strong points in this study.

5. Considering the reader, the Conclusion section is also too long and should be more concise. In lines 566-568, this is an oversimplification, as the authors did not examine the involvement of the transporters. Since this is a discussion/consideration and not the result of the present study, this should not be listed in the Conclusion section.

6. The report by Mishima et al (1995) is an important paper for understanding the interaction between Cd and zinc. However, at that time in 1995, it was not yet known that ZIP8 was involved in the transport of Cd, i.e., the paper alone is an old consideration for understanding the mechanism of Cd-Zn interaction.

7. line 25: Cadmio is Italian and written as “Cd” in English.

Author Response

  1. The reason for using MTF-1KO cells in the Cd-Zn interaction should be described in more detail. The author set out to compare among the contribution of MTs, GSH, and Cd accumulation in the defense of Zn against Cd toxicity in the fibroblasts suppressed the synthesis of MTs by knocked out MTF-1? If so, it is essential to examine the expression Cd transporter (e.g. ZIP8).

Reply:

We have replaced the last paragraph with the following:

“Given the important role of MTF-1 in regulating several genes involved in metal homeostasis, we wanted to examine its role in the defense against cadmium-induced toxicity. Since MTF-1 is also sensitive to zinc concentrations, we set up experiments in which cells were co-treated with both metals. The use of a null cell strain for MTF-1 should highlight the role played by this factor in response to cadmium-induced toxicity and related defense mechanisms. In future studies, the expression of the ZIP8 transporter will be examined, but also of others, such as ZnT1.

  1. I interpreted “ZnCd 1” to mean “50 µM Zn and 2 µM Cd.” However, the such abbreviations should be mentioned in the manuscript and in the figures, respectively. The authors should describe them so that the reader does not misunderstand.

Reply:

Thank you for the comment.

I have mentioned the abbreviations in the manuscript (both in the results and in the materials and methods) and in the figures.

  1. In Figure 1, it is common to describe it as “Viability” instead of “Vitality”. In addition, these data must be analyzed for statistical significance. In particular, as in Table 1, three major points should be analyzed: First, in order to clarify the effect of zinc on the Cd toxicity, the significant difference between the viability of Cd1 and ZnCd1, Cd2 and ZnCd2 in WT should be analyzed. Second, to clarify the effect of MTF-1 KO on its Cd toxicity, the significant difference between the viability of Cd 1 in WT and MTF-1 KO should be analyzed. Third, the significant difference between the viability of CdZn1 and CdZn2 in WT and MTF-1 KO should be analyzed to clarify the effect of MTF-1 KO on the Cd-Zn interaction. The data should also be redesigned to facilitate the understanding of these analyses. The markers in the figures are all circle-like, and there is no uniformity in the colors of the markers and lines, which will confuse the reader as it is.

Reply:

We substituted Vitality with viability.

We changed Figure 1, and now the significances are reported and described in the results.

  1. Since it is difficult to understand what was obtained in the present study, the structure of the entire text in Discussion section should be reconsidered. In particular, it should be clearly stated whether zinc protects against Cd toxicity, what the mechanism is, and what was obtained by using MTF-1 KO. Currently, the present study only compares with previous papers, and although the validity of each and every data is discussed, it is difficult to find strong points in this study.

Reply:

We thank the reviewer for the suggestions. We reviewed the discussion, trying to highlight the role of zinc against Cd toxicity and the results obtained utilizing MTF-1 KO

  1. Considering the reader, the Conclusion section is also too long and should be more concise. In lines 566-568, this is an oversimplification, as the authors did not examine the involvement of the transporters. Since this is a discussion/consideration and not the result of the present study, this should not be listed in the Conclusion section.

Reply:

We reviewed and shortened the conclusion section.

  1. The report by Mishima et al (1995) is an important paper for understanding the interaction between Cd and zinc. However, at that time in 1995, it was not yet known that ZIP8 was involved in the transport of Cd, i.e., the paper alone is an old consideration for understanding the mechanism of Cd-Zn interaction.

Reply:

It is true what the reviewer observes, in fact we also cite the work of Zhang et al. (2014) and He et al. (2006)

  1. line 25: Cadmio is Italian and written as “Cd” in English.

We have corrected cadmio with cadmium

Reviewer 3 Report

In this manuscript, the authors investigated the effects of Cd on cytotoxicity in a fibroblast cell line corresponding to an SV40-transformed MTF-1 mutant (MTF1-/-) and in wild type (MTF1+/+) as well as the role of Zn in cellular protection. 

The aim of co-treatment in knock-out lines is to clarify the mechanism of action of Zn. The scientific work is very interesting, however, some problems, as indicated below, should be addressed before the document can be considered for the publication. This version of the manuscript is not enough complete.

The paper is clearly written and complete in most respects. English language and style are fine, minor spell check required. In general, I suggest to review the style of the manuscript according to the guidelines of the journal.

The authors should modify the abbreviation "Zn" and "Cd". This version is not correct.

In the abstract (section Methods) the incubation times as well as the concentrations of the heavy metals should be added.

I suggest to add this reference in the Introduction section (DOI:10.4081/jbr.2018.7203). The authors should provide more knowledges about the action of this heavy metal in nucleated and anuclated cells, namely erythrocytes. Heavy metals find their way through blood stream allowing their distribution to tissues and organs. Among blood cells, human erythrocytes (RBC) are an important target of toxicity because this metal ion preferentially accumulates in these cells, reaching concentrations higher than those found in plasma.

Could used concentrations be accumulated in this cell line? In which conditions? In addition, fibroblast cell line is used as a model to invetsigate action of heavy metals: why?The authors should explain better this aspect.

The authors should investigate the ratio between GSH/GSSG

Author Response

We thank the reviewer for the suggestions. We revised the manuscript according to the guidelines of the journal

In the abstract (section Methods) the incubation times as well as the concentrations of the heavy metals should be added.

Reply:

Thank you. We have added the incubation time and the concentrations of heavy metals in the abstract.

I suggest to add this reference in the Introduction section (DOI:10.4081/jbr.2018.7203). The authors should provide more knowledges about the action of this heavy metal in nucleated and anuclated cells, namely erythrocytes. Heavy metals find their way through blood stream allowing their distribution to tissues and organs. Among blood cells, human erythrocytes (RBC) are an important target of toxicity because this metal ion preferentially accumulates in these cells, reaching concentrations higher than those found in plasma.

Reply:

We added the reference suggested by referee in the Introduction section.

The authors should investigate the ratio between GSH/GSSG

Reply:

The reviewer is right: one of the points to be investigated in future research will be the relationship between GSH/GSSG

Round 2

Reviewer 1 Report

I think the present version has been largely improved. Having said this I still would like the authors to overhaul their discussion. It is too long and detailed and the internal structure is not obvious for the non-expert reader. Maybe they can describe the mechanism of metal detoxification and then work in how their data support that mechanisms, and how GSH/GSSG is important in that respect. While the discussion is better structured, too many aspects and technical details are described and therefore the focus is lost. It would really need an effort, not just deleting 2-3 sentences. I just think the paper would really benefit, because the data are very interesting.

Author Response

Thanks for the suggestions.
We revised the discussion further following the reviewer's advice. We inserted a sentence at the beginning of the discussion in relation to metal toxicity and detoxification mechanisms that we will later discuss. In the first part of the discussion we relate the accumulation of metals to transporters. The relationship with the GSH and the ratio GSH/GSSG follows. Finally we discuss the results related to MTs. In all of this the possible role of MTF-1 is discussed in the light of the results obtained with the null strain. We have eliminated some technical details. We hope that the manuscript will now be accepted for publication.

Reviewer 2 Report

The revised manuscript provides better content to understand this topic.

One point: In lines 77-78, "In future studies, ~" should be at the end of the discussion, not in the introduction, since it is a future perspective.

Author Response

Thanks for the suggestion.
We have moved the sentence from the introduction to the discussion, to the end of the discussion of the transporters.